# Initial Experience with Single-Session Resin-Based Transarterial Radioembolization Mapping and Treatment of Small Hepatocellular Carcinomas

**DOI:** 10.3390/cancers17081265

**Published:** 2025-04-09

**Authors:** Michael Mohnasky, Sandra Gad, Marco Fanous, Johannes L. Du Pisanie, Marija Ivanovic, David M. Mauro, Hyeon Yu, Alex Villalobos, Andrew M. Moon, Hanna K. Sanoff, Jingquan Jia, Nima Kokabi

**Affiliations:** 1School of Medicine, University of North Carolina at Chapel Hill, Chapel Hill, NC 27599, USA; sandra_gad@med.unc.edu (S.G.); marco_fanous@med.unc.edu (M.F.); 2School of Medicine, Saint George’s University, West Indies P.O. Box 7, Grenada; 3Department of Radiology, School of Medicine, University of North Carolina at Chapel Hill, Chapel Hill, NC 27599, USA; lourens_dupisanie@med.unc.edu (J.L.D.P.); marija.ivanovic@unchealth.unc.edu (M.I.); david_mauro@med.unc.edu (D.M.M.); hyeon_yu@med.unc.edu (H.Y.); nima_kokabi@med.unc.edu (N.K.); 4Department of Gastroenterology and Hepatology, School of Medicine, University of North Carolina at Chapel Hill, Chapel Hill, NC 27599, USA; andrew.moon@unchealth.unc.edu; 5Lineberger Comprehensive Cancer Center, University of North Carolina at Chapel Hill, Chapel Hill, NC 27599, USA; 6Department of Medicine, School of Medicine, University of North Carolina at Chapel Hill, Chapel Hill, NC 27599, USA; hanna_sanoff@med.unc.edu (H.K.S.); jingquan_jia@med.unc.edu (J.J.)

**Keywords:** transarterial radioembolization, hepatocellular carcinoma, interventional oncology, ^99M^technetium macroaggregated albumin, resin microspheres

## Abstract

Yttrium 90 (Y90) transarterial radioembolization is a cancer treatment for liver cancer, namely, hepatocellular carcinoma. It is most commonly performed over two treatment days, where liver mapping is performed on the first day and then the radioactive Y90 is delivered during the second session. New research has shown that select patients may be able to safely forgo the initial mapping session and have their treatment streamlined to just a single day. This study examined the safety and efficacy of a single-session mapping and treatment in a small group of individuals. We found that patients achieved a good response without any serious complications, which concurs with recent reports in the literature, and indicates that this new streamlined treatment design could be widely adopted. If so, patients would undergo fewer procedures, with the associated logistical and clinical implications, while achieving the same outcomes.

## 1. Introduction

Liver cancer, of which hepatocellular carcinoma (HCC) makes up 75%, is the sixth most common cancer worldwide and is the third leading cause of cancer-related deaths [1]. The 2022 Barcelona Clinic Liver Cancer (BCLC) prognosis and treatment strategy endorses Yttrium 90 (Y90) transarterial radioembolization (TARE) as a potential treatment option for very-early (BCLC 0) and early-stage (BCLC-A) HCC based on the landmark LEGACY study [2,3]. Conventionally, Y90 TARE treatment is performed over two sessions. The initial session is composed of mapping angiography to delineate the tumor vascular supply and safe treatment location by technetium-99m macroaggregated albumin (^99m^Tc-MAA) injection intra-arterially. The distribution of ^99m^Tc-MAA is then measured by single-photon emission computed tomography (SPECT), which allows interventionalists to calculate the correct tumoricidal dose based on the volume and degree of activity distribution. Additionally, this allows for the calculation of hepatic shunting, notably to the lungs (i.e., the lung shunt fraction (LSF)), the gastrointestinal (GI) tract, and elsewhere to prevent off-target radiation injury. Patients are then discharged and return for the second session, which is frequently scheduled 2–3 weeks later. During this second session, patients undergo repeat angiography, and the prescribed activity of Y90 microspheres are injected into the pre-planned targeted vessel(s) for treatment [4].

To prevent radiation pneumonitis secondary to non-target particle shunting, the current standard of care is for all patients to undergo mapping and LSF calculation. Historically, an LSF > 20% was used as a contraindication for patients undergoing Y90 TARE [4,5]. However, the consensus guidelines for both glass and resin now recommend using the projected lung dose of >30 Gy as an exclusion criterion for treatment [6,7]. Performing mapping and treatment in two separate sessions meaningfully increases the burden on patients and healthcare systems from the added travel and missed work, procedure time, and sedation/anesthesia cost. The need for a second procedure may increase patient risk from arterial access and intervention twice, as well as additional radiation and contrast exposure. The need for pre-treatment mapping has been recently called into question by a large retrospective study of undergoing pre-TARE mapping. Among the 448 patients meeting the Organ Procurement and Transplant Network (OPTN) T1/T2 criteria (single HCC <5 cm or ≤3 nodules all ≤3 cm) without evidence of macrovascular tumor invasion (MVI) or a history of transjugular intrahepatic portosystemic shunts (TIPS), all had an LSF <18%, suggesting that single-session mapping and treatment (SSMT) approaches may be safe and feasible for well-selected patients [8].

This study aimed to retrospectively evaluate the initial safety, efficacy, and impact on time to treatment for a Y90 SSMT approach in OPTN stage T2 HCC treated specifically with resin microspheres using intraprocedural personalized dosimetry planning.

## 2. Materials and Methods

### 2.1. Eligibility Criteria

Retrospective review was conducted for all patients who underwent SSMT Y90 TARE treatment with resin microspheres at this institution between September 2023 and May 2024. Patients that were OPTN stage T2 HCC without MVI or TIPS were deemed eligible for this treatment approach. Patients were selected to undergo the SSMT approach based on eligibility criteria and at the clinical discretion of the multidisciplinary liver cancer tumor board and treating physicians, including the evaluation of factors such as the likelihood of benefit, likelihood of successful targeting, history of liver-directed therapies, etc. This study was approved by this institution’s institutional review board.

### 2.2. Treatment Approach

For the single-session approach, patients were treated by a single interventional radiologist with years of experience in resin-based Y90 therapy. After the angiographic selection of the hepatic tumoral vasculature, the perfused segment/angiosome volume was calculated using cone-beam computed tomography (CBCT), and a personalized single compartment dose of ≥250 Gy to the segment was prescribed using the medical internal radiation dosimetry (MIRD) model [9]. While the target vessel remained catheterized, nuclear medicine and radiopharmacy prepared the personalized dose and transported the 4- or 5-day pre-calibrated Y90 microspheres to interventional radiology (IR). The dose was then delivered at the planned location, followed by catheter removal and hemostasis. Following injection, patients underwent SPECT/CT imaging to ensure the appropriate targeting of the tumor and to exclude obvious non-target embolization. Post-Y90 dosimetry was performed using MIM Sureplan^®^ (MIM, Cleveland, OH, USA) (Figure 1). 

LSF and lung dose could not be accurately calculated due to the inherent limitations of post-Y90 TARE bremsstrahlung SPECT imaging, as well as the post-processing image reconstruction software, so these variables were not assessed.

### 2.3. Outcomes

Follow-up magnetic resonance (MR) imaging was obtained at 8–12 weeks post-treatment, in concordance with standard institutional protocols. Treatment response on imaging was determined via the mRECIST criteria by the treating interventional radiologists. Adverse events (AEs) ≥ Grade 3 according to the Common Terminology for Adverse Events v5.0 (CTCAE v5.0) were assessed via a retrospective review of the electronic medical record from treatment to the 12-week post-TARE clinic visit. Clinical success was defined as complete response per the mRECIST criteria on follow-up imaging and the lack of development of extra-hepatic radiation-induced sequalae within 30 days post-procedure.

### 2.4. Comparison Group

Patients undergoing the SSMT treatment approach were compared to patients who underwent the traditional treatment approach. All of the patients who underwent standard Y90 mapping and treatment with glass microspheres between May 2022 and May 2024 were included as a comparison group. Glass microspheres were chosen as the comparison group, given that individuals at this institution using resin microspheres were participating in clinical trials undergoing nonconventional mapping and treatment strategies. At this institution, providers use either glass or resin microspheres for all of their patients based on personal preference, and no difference in indication exists in which one patient receives one modality over the other. Days from IR clinic visit to Y90 treatment, total procedure time, and total radiation dose were recorded for both groups. The procedure time and fluoroscopy time for the traditional treatment group are the sum of both the mapping and treatment sessions. Time from the IR clinic visit to the Y90 treatment was determined by the number of days between a documented clinic visit for the purpose of the Y90 evaluation to the day of treatment. Because of the confounding introduced by the different indications for SSMT vs. traditional TARE, the efficacy and safety outcomes were not compared between groups.

### 2.5. Cost Analysis

The cost associated with each treatment approach was estimated via Center of Medicare Services (CMS) 2025 reimbursement codes for the various components of each treatment algorithm. Hospital outpatient costs in addition to physician services fees were added together for the typical components of a mapping and Y90 treatment procedure to provide a general estimate of the total costs associated with the mapping and treatment portions. Because this was a general cost estimate, other factors that contribute to the overall cost to the healthcare system and the patient, such as the procedure time, total equipment used, anesthesia time, etc., were not included.

### 2.6. Statistical Analysis

Statistical analyses comparing the SSMT treatment group with the traditional treatment group were performed using the Shapiro–Wilk test for normality and the Mann–Whitney U test. All of the statistical analyses were performed using SPSS Statistics v29.0.2.0 (IBM Corp., Armonk, NY, USA).

## 3. Results

Twelve SSMT treatment sessions were conducted on 10 consecutive patients during the study interval. Two patients received two separate SSMT treatments performed at different time points; one patient received two treatments for tumors in different segments, while the other patient received an additional treatment for the same tumor 28 days after the initial treatment due to the incomplete targeting of the tumor, which was noted on the Y90 SPECT/CT. A total of 60 patients underwent the traditional Y90 treatment approach with glass microspheres during the comparison time interval. Two patients were deemed lost to follow-up between the initial clinic visit and the eventual treatment, so were excluded from the analysis. Patients were younger in the SSMT cohort, with a median age of 64 vs. 69 years in the traditional cohort (*p* < 0.001). Each treatment targeted an isolated lesion with median size of 2.5 cm in the SSMT cohort compared to 3.3 cm (*p* < 0.0001) in the traditional cohort. All of the patients in the SSMT were OPTN stage T2 and below at the time of treatment, whereas 18 patients were stage T3 and 2 patients were stage T4 in the traditional cohort (*p* = 0.48). Similarly, patients in the SSMT cohort were all BCLC stage A and below, whereas 18 patients were BCLC B and 3 were BCLC C (*p* = 0.0054) (Table 1).

The median delivered tumor dose was 377.7 Gy (IQR: 273.5, 570.1 Gy). No ≥Grade 3 AEs per CTCAE v5.0 were experienced in the immediate post-procedure setting, as well as by the time of the follow-up clinic visit at 12 weeks post-Y90. No patients developed radiation-induced pneumonitis or other radiation-induced sequelae within 12 weeks. Follow-up MR imaging with a median follow-up of 79 days (IQR: 58, 86) demonstrated complete response per mRECIST in 11/12 treatments. One patient had a partial response that improved to complete response after re-treatment (Table 2). During the total retrospective follow-up period from the first SSMT to the analysis (402 days), one patient died due to liver failure at day 259 post-treatment, which was not clinically attributed to the TARE.

The median time from the initial IR clinic visit for the TARE evaluation to the TARE treatment was lower for the SSMT group (26.5 vs. 61 days; *p* < 0.001). Both the median procedure time (142 vs. 151 min; *p* = 0.12) and median fluoroscopy time (25.1 vs. 22.9 min; *p* = 0.28) were similar for both the SSMT and traditional treatment groups. These findings persisted when the traditional treatment cohort included only patients meeting SSMT criteria (N = 39) (Table 3).

The cost of an SSMT session was estimated to be USD 37,969.20 (Y90 treatment cost) compared to USD 64,331.31 (mapping cost combined with Y90 treatment cost) for a traditional treatment approach (Table 4).

## 4. Discussion

This investigation adds to the recent literature demonstrating that SSMT Y90 treatment strategies in select patients with small HCCs achieve high efficacy and safety. Berman et al. recently published a retrospective study of 15 patients utilizing this SSMT approach with glass microspheres and reported a 94% complete response rate, while inducing no cases of radiation pneumonitis [10]. A subsequent study of 62 patients undergoing SSMT with glass or resin microspheres achieved a response rate and safety profile equivalent to traditionally treated patients [11]. This study additionally showed that patients with small HCCs undergoing SSMT Y90 TARE treatment protocols have comparable cumulative procedure and fluoroscopy times as patients treated via the standard two-session approach. However, the single-session treatment facilitated a significantly shorter interval between initial clinic visit to treatment.

No patients in this study reported sequelae of extra-hepatic radiation toxicity following treatment, which is consistent with the aforementioned studies [10,11]. Together, these results provide clinical support that bypassing the mapping step in patients meeting specific criteria can be performed safely. However, radiation pneumonitis is a rare complication of TARE, and the small sample sizes of this and other previously published studies like it make it difficult to assess if this approach increases the relative risk given this low baseline risk [12,13].

In this study, 11/12 (92%) treatments resulted in a complete response after the initial treatment, which is comparable to the complete response rates for resin reported elsewhere in the literature [3,14]. The interventional radiologist who performed these procedures had years of experience in resin-based Y90 therapy for HCC, which may lead to an overestimation of the therapy success compared to when this approach is performed by a less experienced operator. The mapping session and post-^99m^Tc-MAA imaging provides helpful information regarding tumoral vasculature, in addition to the information gained by arterial interrogation and CBCT during the Y90 treatment. A less experienced operator relying solely on the information gained during the treatment session may be more likely to incompletely target a lesion and, therefore, this approach may be more prone to such an occurrence. However, with careful and systematic interrogation, combined with the confidence of coverage provided by CBCT, it is presumed that the risk of incomplete targeting is not substantially greater than what already exists at baseline. Future studies should therefore examine the relative rate of incomplete treatment to inform whether additional safeguards in the SSMT approach are needed to mitigate this risk. Additionally, when a tumor is grossly undertreated, this is readily apparent on post-treatment SPECT/CT imaging. While there is a theoretical risk of incomplete treatment in the SSMT approach, such an occurrence can be remedied by an additional treatment that may occur within weeks of the initial treatment. Though not ideal, this situation mimics what is typical of the traditional treatment approach (i.e., two sessions), and therefore is not presumed to place patients at a substantially increased risk of progression related to delayed treatment time.

A target segment dose of >250 Gy was used based on previous studies that demonstrated a 95% complete response rate in radiation segmentectomies using resin Y90 [9,14]. These studies also reported that a TD of >337 Gy had an 83% sensitivity for complete response with resin microspheres [9]. While five tumors in this cohort received a TD <337 Gy, all of them achieved complete response per mRECIST.

Time from the IR clinic visit to the treatment was shorter in the SSMT cohort compared to the traditional treatment cohort, and this effect persisted even when those with a higher burden of disease (OPTN stage T3 and T4) were excluded. Logistical difficulties with scheduling both a mapping and treatment session in short succession, in addition to the days elapsing between sessions, led to a difference of 34 days between cohorts. While the time to treatment was lower with the SSMT approach, the total procedure time and total fluoroscopy time did not differ between the approaches, which may be explained by several of the following factors: (1) due to the novelty of this approach, the operators may have spent more time achieving adequate catheter position and ensuring safe delivery; (2) the time from the segment volume calculation to the dose delivery in the SSMT cohort increases procedure time; (3) some operators in the traditional cohort did not utilize cone-beam CT, which could lower the procedure time as a whole.

The cost of an SSMT treatment was estimated to be less than a traditional treatment by USD 26,362.11, i.e., the cost of the mapping session. This cost estimation is not meant to be a specific valuation, given that many other factors that could greatly influence cost, such as the procedure time, anesthesia, and specific equipment used were not thoroughly evaluated. Additionally, costs were estimated using CMS codes, and patients insured by other means would incur a different cost. Such a comprehensive evaluation necessary to confidently elucidate the cost gap between treatment approaches is beyond the scope of this work. However, this estimation does serve to elucidate the presumed cost savings by reducing a two-procedure treatment algorithm to one. This is in addition to the savings of the individual patient by reducing patient-specific costs (i.e., travel, lodging, time off of work, etc.).

While an SSMT approach likely reduces healthcare costs, this approach may also be more feasible for select patients. Patients with logistical challenges to attending multiple treatments or clinical conditions that would favor fewer arterial accesses (i.e., hemophilia) may benefit from an SSMT approach. Research has also found that patients with HCC consider variables such as the treatment time and procedure financial burden when making treatment decisions [15]. Offering a single-session treatment may make TARE a more attractive option for these patients, thereby expanding its reach.

Several important limitations exist and warrant discussion. The retrospective design of this analysis leads to the high potential for error and confounding bias, which may limit the ability to adequately compare these two cohorts. Differences in operator techniques and the use of glass vs. resin in the SSMT vs. traditional cohorts, respectively, also limit direct comparison, but likely to a smaller degree. The single-institution design limits the generalizability of the results. Finally, the treatment cohort was small, limiting the ability to adequately comment on the safety of such an approach. Larger, prospective studies will be best suited for assessing this outcome and are needed to more confidently assess the safety.

## 5. Conclusions

This study demonstrates that patients with OPTN T2 stage HCC may bypass ^99m^Tc-MAA mapping when treated with resin Y90 without sacrificing the treatment efficacy or increasing the risk. The absence of serious side effects provides clinical evidence for the safety of altering the Y90 treatment paradigm for select patients. By doing so, patients experience a faster time from evaluation to treatment, and patients, along with the healthcare system, could be spared an additional procedure and the associated risks, burdens, and costs. Validating this approach with large, prospective studies is essential to provide stronger evidence supporting this novel approach.

## Figures and Tables

**Figure 1 cancers-17-01265-f001:**
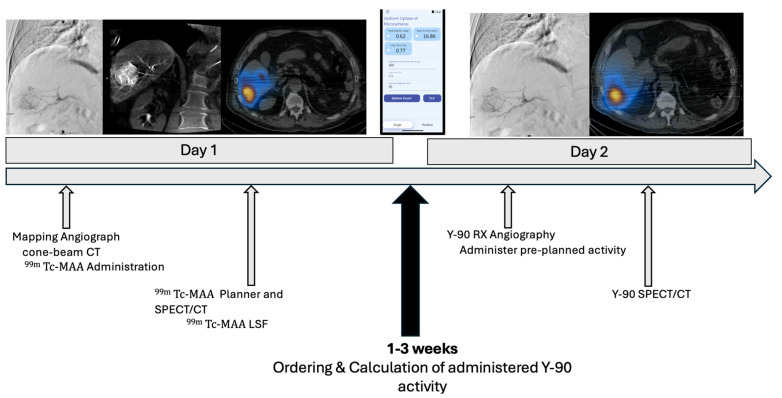
Single-Session Mapping and Treatment Flowchart. Two-tiered flowchart depicting the conventional Y90 TARE treatment paradigm with mapping angiography and ^99m^Tc-MAA administration on day 1, followed by Y90 administration during the second treatment session. The streamlined treatment paradigm is depicted below, with both mapping angiography and Y90 administration on the same day.

**Table 1 cancers-17-01265-t001:** Demographics and clinical characteristics.

Characteristic	Single-Session Mapping and Treatment (N = 10)	Traditional Treatment (N = 58)	*p*-Value
Median age (years), IQR	64 (63, 77)	69 (62, 74)	*p* < 0.001
Male (%)	6 (60%)	51 (88%)	*p* = 0.085
Race (%)			*p* = 0.178
White	8 (80%)	46 (79%)	
Black or African American	1 (10%)	5 (9%)	
Asian	1 (10%)	0 (0%)	
Other	0 (0%)	7 (12%)	
Ethnicity (%)			*p* = <0.001
Not Hispanic or Latino	10 (100%)	53 (91%)	
Hispanic or Latino	0 (0%)	5 (9%)	
Child-Pugh Classification (%)			*p* = 0.24
A	9 (90%)	51 (89%)	
B	1 (10%)	7 (12%)	
Liver-disease etiology			*p* = 0.0023
Alcohol	1 (10%)	10 (18%)	
HCV	4 (40%)	19 (32%)	
MASLD	1 (10%)	14 (25%)	
HCV/Alcohol-associated liver disease	3 (30%)	5 (9%)	
HCV/MASLD	1 (10%)	1 (2%)	
MASLD/Alcohol-associated liver disease	0 (0%)	1 (2%)	
HBV	0 (0%)	2 (4%)	
HCV/Hemochromatosis	0 (0%)	1 (2%)	
Unknown	0 (0%)	5 (9%)	
OPTN Stage			*p* = 0.49
T1	1 (10%)	0 (%)	
T2	9 (90%)	38 (65%)	
T3	0 (0%)	18 (31%)	
T4	0 (0%)	2 (4%)	
BCLC Stage (%)			*p* = 0.0054
0	1 (10%)	4 (7%)	
A	9 (90%)	33 (56%)	
B	0 (0%)	18 (31%)	
C	0 (0%)	3 (5%)	
Median tumor size, IQR (cm)	2.6 (2.1, 2.9)	3.8 (2.7, 6.1)	*p* < 0.0001

**Table 2 cancers-17-01265-t002:** Dose delivered and treatment outcome.

Outcome	*n* = 12
Median tumor dose (Gy), IQR	377.7 (273.5, 570.1)
≥Grade 3 AEs	0 (0%)
Median imaging follow-up (days), IQR	79 (58, 86)
Initial imaging response (*n*= 10)	
Complete response	9 (90%)
Partial response	1 (10%)

**Table 3 cancers-17-01265-t003:** Comparison of the time to procedure and procedure characteristics.

	SSMT Cohort (*n* = 10)	Traditional Cohort (*n* = 58)	*p*-Value (SSMT vs. Traditional Cohort)
Median time from IR clinic visit to treatment (days), IQR	26.5 (15.3, 39.0)	61.0 (48.0, 88.8)	*p* < 0.001
Median procedure time (minutes), IQR	142 (123, 152)	151 (130, 199)	*p* = 0.12
Median fluoroscopy time (minutes), IQR	25.1 (22.0, 29.0)	22.9 (15.6, 31.8)	*p* = 0.28

**Table 4 cancers-17-01265-t004:** Cost comparison of mapping and Y90 treatment procedures.

	Hospital Outpatient (OPPS) (USD)	Physician Services (MFFS) (USD)	Total Cost (USD)
Mapping Total Cost	25,400.86	1231.25	26,632.11
75,726: Angiography, visceral, RS&I	5405.70	90.57	
37,242: Arterial emb or occ, RS&I; arterial other than hem or tumor (includes catheter placement, 3D post-scan, ^99m^TC-MAA dose, radiopharmaceutical quantification measurement)	17,956.72	1042.99	
74,170: CT, abdomen; w and w/o contrast	178.02	63.40	
78,801: Planar imaging of multiple areas	401.83	32.35	
78,832: SPECT/CT imaging of multiple areas	1458.59	92.51	
Y90 Treatment Total Cost	36,685.53	1283.67	37,969.20
75,726: Angiography, visceral, RS&I	5405.70	90.57	
C2616: Brachytherapy source (yttrium-90 non-stranded–Medicare)	17,412.53	-	
37,243: Vascular emb or occ, inclusive of all RS&I, intraprocedural road mapping, and imaging guidance necessary to complete intervention; for tumors	11,340.57	858.64	
77,370: Special Medical Radiation Physics Consultation	132.77	-	
77,470: Special Treatment Procedure	578.47	104.48	
77,300: Basic Dosimetry Calculation	132.77	32.02	
79,445: Radiopharmaceutical therapy, intra-arterial particulate admin (1 doctor model (IR/AU))	224.13	105.45	
78,832: SPECT/CT imaging of multiple areas	1458.59	92.51	

## Data Availability

Data are available upon request.

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
