# Peer review of "Initial Experience with Single-Session Resin-Based Transarterial Radioembolization Mapping and Treatment of Small Hepatocellular Carcinomas"

_cancers, 2025, doi:10.3390/cancers17081265_

Round 1
Reviewer 1 Report
Comments and Suggestions for Authors
It has become increasingly apparent that alcoholism is the primary underlying cause of death throughout the world particularly among patients with steatohepatitis. Non-alcoholic steatohepatitis is a misnomer since the criteria include defined amounts of alcohol depending for men or women. The practical problem is generating an honest history since the ramifications of an alcohol history on patient's job, family, employer, litigation, driver's license and decision for liver transplantation are all possibly jeopardized. Moreover there is very recent date suggesting that any alcohol may predispose the individual to malignancy including HCC. Your group should consider collaborative studies to assess the presence of HCC in alcoholic patients. In Table 1 only 30% HCV/Alcohol-associated liver disease. The quest for early detection of HCC is likely to be lifesaving in many cases.
Author Response
We agree with the reviewers comments on the role of alcohol in liver disease and HCC. We thank you for your insight and appreciate the encouragement to pursue additional studies regarding the link between alcohol and HCC.
Reviewer 2 Report
Comments and Suggestions for Authors
This study aimed to retrospectively explore the safety and efficacy and impact on time to treatment for a Y90 single session mapping and treatment (SSMT) approach in organ procurement and Transplant Network (OPTN) stage T2 HCC treated specifically with resin microspheres using intraprocedural personalized dosimetry planning. Overall, ten consecutive patients were treated with 12 treatments. Complete response was achieved in 11/12 patients (92%). The conventional cohort consisted of 60 patients, all OPTN T2 treated with radiation segmentectomy with glass microspheres. Patients undergoing SSMT had a median time from clinic visit to treatment of 26.5 days (IQR: 15.3, 39) vs. 61 days (IQR: 48, 88.8) in the conventional TARE group (p<0.001). They concluded that streamlined single session resin-based Y90-TARE in patients with OPTN T2 stage HCC is feasible, efficacious, safe and associated with reduced time to treatment.
The study is of interest and of potential clinical significance. However, the authors concluded that patients with OPTN T2 stage HCC may bypass 99mTc MAA mapping when treated with resin Y90 without sacrificing treatment efficacy or increasing risk. The absence of serious side effects provides clinical evidence for the safety of altering the Y90 treatment paradigm for select patients. To draw firm conclusions on treatment safety, the authors should recall and discuss the impact and the importance of liver functional reserve in the treatment of hepatocellular carcinoma. In particular, a very importan issue for the safety profile of any treatment, is the number and type of previous HCC treatment since it has been recently described the prognostic impact of previous HCC treatment in the residual liver function deterioration risk, as recently recommended (J Hepatol. 2022 May;76(5):1185-1198. doi: 10.1016/j.jhep.2021.11.013). There are previous studies demonstrating that repeated transarterial treatments as weel as previous surgical resection might be associated to the higher risk of liver function deterioration with a switch of liver function from Child-Pugh class A to Class B that is associated with higher risk of treatment-related adverse events as well as higher risk of liver function deterioration, as previously described (Lancet Oncol. 2017 Feb;18(2):e101-e112. doi: 10.1016/S1470-2045(16)30569-1.).
Author Response
"To draw firm conclusions on treatment safety, the authors should recall and discuss the impact and the importance of liver functional reserve in the treatment of hepatocellular carcinoma. In particular, a very importan issue for the safety profile of any treatment, is the number and type of previous HCC treatment since it has been recently described the prognostic impact of previous HCC treatment in the residual liver function deterioration risk, as recently recommended"
Thank you for pointing this out. We agree that the amount of functional liver reserve that a patient has entering the therapy is a very important prognostic factor and influences heavily the rate of complications related to liver function. To address this point, we added on line 95 a clarification that the multidisciplinary team considers a variety of factors including history of liver-directed therapies in selecting a specific treatment for a specific patient. Because these patients were selected for this approach clinically and this study is a retrospective evaluation of those patient selections, the patients included in this analysis were already deemed to have a supposed sufficient post-treatment liver capacity. We therefore elected for a general statement further elucidating that point in lieu of outlining the entire decision making process of the multidisciplinary team because 1) the thoughts behind the clinical decision making are unknown to us in this retrospective analysis and 2) outlining a specific history of liver-directed therapies would also warrant then outlining all of the other specific eligibility criteria. We appreciate the attention to this detail and how it improved the methods section of the manuscript.
Reviewer 3 Report
Comments and Suggestions for Authors
Cancers. Title: Initial Experience with Single-Session Resin-Based Transarterial Radioembolization Mapping and Treatment of Small Hepatocellular Carcinomas.
- What were the differences between the clinical and imaging characteristics of patients who underwent SSMT and traditional treatment? Did the authors consider doing a propensity score matching?
- In the traditional cohort who decided the time from IR visit to treatment?
- Short follow-up, limits the assessment of outcomes over the long term
- Did the authors calculate differences in cost and quality of life?
- The authors have rightly mentioned small sample size is a major limitation and larger sample size studies are required
Comments on the Quality of English Language
Minor edits required
Author Response
- What were the differences between the clinical and imaging characteristics of patients who underwent SSMT and traditional treatment? Did the authors consider doing a propensity score matching?
Thank you for this point. We agree that this is useful information and have therefore added this to the manuscript. Table 1 now includes a comparison of baseline demographic and HCC characteristics. To be somewhat expected, it is now clear that those in the traditional cohort were older, had larger tumors, and were higher stage. Because patients in the SSMT cohort were selected based on having a smaller burden of disease and the comparison group did not have this restriction, these factors differed as expected. We did not pursue a propensity score matching though we understand that this improves the comparison when a group is small (such as our SSMT cohort). The main intent of the comparison group primarily is to elucidate the typical timeframe for a Y90 evaluation to eventual treatment. Having a larger group for this purpose helps to demonstrate the wide range of times that are found in the real-world clinical setting so we opted to not do a propensity score to keep the typical range as realistic as possible.
- In the traditional cohort who decided the time from IR visit to treatment?
Thank you for this point. We added on line 136 that time from IR visit to treatment was determined by what is documented in the medical record from day 0 as time of clinic visit for Y90 evaluation to the day when Y90 was administered.
- Short follow-up, limits the assessment of outcomes over the long term
We agree with this point and agree that it is a limitation of the study design. We hope that further studies including prospective evaluations would mitigate this issue.
- Did the authors calculate differences in cost and quality of life?
Thank you for this comment. Yes, we decided to estimate the cost of the two treatment algorithms based on CMS reimbursement which is now included as a new Table 4 with associated discussion of the results. Because this work is not primarily a cost-analysis and the very specific costs are difficult to estimate based in a retrospective manner, a simple estimation was done. The ways that this treatment algorithm impacts patient quality of life can be supposed and discussed in the manuscript between lines 300-321. This is, however, impossible to retrospectively estimate unlike the general procedure costs.
- The authors have rightly mentioned small sample size is a major limitation and larger sample size studies are required
We agree that this is a limitation of the manuscript and hope to remedy this is future studies.
Round 2
Reviewer 3 Report
Comments and Suggestions for Authors
The authors have made changes as suggested.